# Score-Based Generative Models for Binding Peptide Backbones

**John D Boom** [* 1]   **Matthew Greenig** [* 2]   **Pietro Sormanni** [2]   **Pietro Liò** [3]

## Abstract

Score-based generative models (SGMs) have emerged as powerful tools for protein design, capable of generating protein structures for a variety of biologically relevant design specifications. Among these, the ability to generate structures capable of binding a specified target holds particular relevance for a range of applications. Despite the success of SGMs in this domain, there has been little systematic exploration of the impact of model design choices for protein binder backbone generation, in part due to the lack of appropriate metrics for generated backbones and their complementarity to the target protein. Here we present LoopGen, a flexible SGM framework for the generation and evaluation of *de novo* binding protein backbones in the absence of inverse folding/folding models. This decoupling from existing inverse folding/folding models not only provides an orthogonal set of metrics but also enables the evaluation of protein structure SGMs in domains where such models are difficult to obtain (e.g. peptide design). We apply our framework to design antibody CDR loop structures, a class of peptides with notable structural diversity, and evaluate a variety of model design choices, showing that choices of structural representation and variance schedule have dramatic impacts on model performance. Furthermore, we propose three novel metrics for testing the dependency of a generated binder structure on its target protein, and demonstrate that LoopGen's generated backbones are indeed conditioned on the sequence, structure, and position of its input epitope. Our results identify promising avenues for further development of SGMs for protein design.

## 1. Introduction

Score-Based Generative Models (SGMs) have proven to be powerful tools for designing novel proteins (Ingraham et al., 2023; Watson et al., 2023; Yim et al., 2023; Bennett et al., 2024; Luo et al., 2022; Anand & Achim, 2022), and the capacity of these models to generate *binding* proteins for pre-specified targets has attracted considerable interest. Due primarily to difficulties adapting SGMs to discrete data modalities, all experimentally-validated deep learning-based *de novo* protein design methods to-date have relied on a two-step process to generate novel proteins (Ingraham et al., 2023; Watson et al., 2023; Bennett et al., 2024). First, an SGM generates an "unlabelled" protein backbone - parameterised in some continuous space - and second, an inverse folding model is used to conditionally generate a sequence that is likely to fold into that structure. Evaluating models that generate protein backbones without sequences is highly non-trivial due to the diversity of both natural protein structures and the synthetic structures that a generative model is expected to produce. The evaluation of generated backbones in the context of binder design is even more challenging, since the extent to which the designed backbone must be constrained by the target protein is often unclear.

The standard metric for evaluating the suitability of generated protein backbones for protein design tasks is *designability*, which attempts to measure the number of unique sequences that fold into the generated backbone structure. In natural proteins, this property correlates with mutational robustness and thermostability (Helling et al., 2001). Typically, designability is measured by sampling sequences for a designed backbone with an inverse folding model (e.g. Dauparas et al. (2022)), re-folding the generated sequences with a folding model (e.g. Jumper et al. (2021)), and calculating the RMSD between the re-folded structures and the original generated sample (Yim et al., 2023; Gaujac et al., 2024). Key limitations of this approach are that it couples the evaluation of the generative model to existing inverse folding/folding models and that it does not necessarily provide any information about the generated structure's functional features, which are particularly relevant in the conditional design setting. Furthermore, designability calculations rely on having access to effective inverse folding and folding models, hindering their application in settings where such models do not perform well or are difficult to

---
[*]Equal contribution  [1]Department of Engineering, University of Cambridge, Cambridge, UK [2]Department of Chemistry, University of Cambridge, Cambridge, UK [3]Department of Computer Science, University of Cambridge, Cambridge, UK. Correspondence to: John D Boom <jb5005@cumc.columbia.edu>, Matthew Greenig <mg989g@cam.ac.uk>.

*Accepted at the 1st Machine Learning for Life and Material Sciences Workshop at ICML 2024.* Copyright 2024 by the author(s).

obtain.

While a variety of SGM-based approaches for protein binder design have been proposed (Watson et al., 2023; Ingraham et al., 2023; Bennett et al., 2024; Luo et al., 2022; Xie et al., 2023), differences in data curation, model architecture, and evaluation methodology between works have made controlled comparisons of model design choices challenging thus far, further exacerbated by the fact that standard metrics for evaluating designed binders *in silico* have not been established. Therefore, we sought to develop a framework for training and evaluating different SGMs in the context of an important task: generating binding loop structures in antibodies, a key class of biomolecules widely applied as therapeutics, diagnostics, and research tools (Sormanni et al., 2018; Lu et al., 2020). Antibody binding is mediated primarily through interactions between the target and short loop regions called complementarity-determining-regions (CDRs), which often lack secondary structure and can exhibit extreme sequence and structural diversity (Fernández-Quintero et al., 2020), posing challenges in the generative modelling setting. Designing CDR structures for binding is also challenging due to limited data availability; around 8000 total antibody structures are available in the PDB, many of which are redundant or lack a binding partner (Dunbar et al., 2014). These challenges, combined with the general utility of antibodies (Sormanni et al., 2018; Lu et al., 2020), motivate the development of novel methods for generating and evaluating CDR structures *in silico*.

Our main contributions are the following:

1. We formulate a flexible, architecture-agnostic SGM that allows rotational information to be straightforwardly added/removed from the model, providing the first controlled comparison, to our knowledge, of protein diffusion models trained on residue orientations and coordinates versus coordinates alone. We demonstrate that reasoning over rotations is essential for generating diverse CDR loop structures.

2. We investigate the use of different variance schedules for coordinates and rotations, observing patterns that motivate the development of new schedules.

3. We identify that ground-truth RMSD, the most common metric in assessing generative models of CDRs, surprisingly does not correlate with physicochemical violations in designed backbones.

4. Based on the limitations of RMSD, we introduce a set of metrics that evaluate the model's ability to generate backbones conditional on a target protein.

5. We release a modular codebase, model weights, and curated CDR loop dataset to enable further exploration.

## 2. Methodology

Score-based generative models on the manifold $SE(3)$ have been studied extensively in the context of protein design (Yim et al., 2023; Watson et al., 2023) as a means of generating full-atom backbone structures. These models constrain the structural solution space to a single rotational and translational component ("frame") per-residue, reducing physicochemical violations by fixing backbone bond lengths and angles to constant values. Despite these benefits, $SE(3)$ generative models introduce unique technical challenges primarily due to the non-euclidean topological structure of the Riemannian manifold of proper 3-D rotations, $SO(3)$. Generation of translations alone is more straightforwardly defined within standard SGM frameworks, and hence many generative modelling frameworks for proteins have focused solely on generating $C\alpha$ atom positions (Lin & AlQuraishi, 2023; Trippe et al., 2023), which often serve as reasonable structural templates. In the context of peptide binders, and particularly binding loops, we hypothesised that generative modelling of frames would provide unique benefits due to the lack of secondary structure associated with loop regions, since accurate reconstruction of the entire backbone from $C\alpha$ positions often relies on secondary structure modelling (Badaczewska-Dawid et al., 2020). Hence we sought to conduct a controlled comparison of generative models for binding loop design, comparing SGM approaches that operate on frames to those that model coordinates only. We combine these two modalities into an SGM framework known as LoopGen.

LoopGen is a deep learning tool and SGM framework for the generation of binding peptides which facilitates direct comparison of model design choices, such as structural representation (e.g. "frames" versus coordinates alone), different estimator architectures, and different choices of variance schedule (see Appendix for more details). We apply LoopGen to generate antibody CDR loops conditioned on a target epitope. For our experiments, we use a heterogeneous variant of the Geometric Vector Perceptron (GVP) GNN architecture (Jing et al., 2021) to estimate scores for CDR residues in each CDR/epitope complex. Since this architecture can flexibly reason over both frames and coordinates alone by adding/removing orientational information via type-1 (vector) features, we deemed it an appropriate choice to conduct a controlled comparison between a frame-based generative model and its $C\alpha$-only counterpart. Each complex is represented as a heterogeneous graph with edges drawn between each residue and its $K = 6$ nearest neighbors in both the CDR and epitope. No sequence information for the CDR is provided. In every model, epitope residues are represented using node features describing their sequence identity and backbone geometry. Self-conditioning is performed as in Watson et al. (2023) at a rate of 0.5. All results were obtained with a

4.3 million parameter model that was trained in 30 hours on one NVIDIA RTX 8000 GPU. Code, model weights, and training data for LoopGen are publicly available at `https://github.com/mgreenig/loopgen`.

Training was first conducted using a large dataset of CDR-like fragments obtained from the PDB90 database (Aguilar Rangel et al., 2022), and subsequent finetuning was performed using a smaller, higher-fidelity dataset (SAbDab) of real CDRs in complex with epitopes (Dunbar et al., 2014), filtering for structures with <90% sequence identity. SAbDab has been used in previous generative experiments (Luo et al., 2022); however, we observed that many of the antibody sequences in SAbDab are redundant: 72% of SAbDab antibodies had >90% sequence similarity to another in the set. Therefore, we curated a non-redundant version of CDRs in SAbDab and removed any antibodies with >90% sequence similarity to minimize train-test leakage. Generation of novel loop structures was performed for 687 test set CDR/epitope complexes from SAbDab, generating ten loops conditioned on the epitope structure and with the same length and centre of mass as the ground truth CDR. Furthermore, to evaluate the generative model's dependence on epitope information, we generate ten CDR loops for three transformed versions of each epitope in the test set: permuted, sequence scrambled, and translated (see Appendix for more details).

## 3. Experiments

### 3.1. Evaluating the Impact of Frames

We first investigated the effect of incorporating rotational information in the diffusion process. We compared sets of CDR structures produced by generative models of rotations and positions (frames) versus models of C$\alpha$ coordinates alone. Interestingly, the C$\alpha$ RMSD[1] of the generated structures compared to the ground truth was indistinguishable between the models; however, structures generated by the coordinates-only model deviated more significantly from the ground truth distances between adjacent C$\alpha$ atoms in the backbone (Fig. 1, middle). To analyze the diversity of the generated structures, we computed the mean pairwise RMSD (mpRMSD) between the 10 generated CDRs structures against each test set epitope for both models. Structures generated by the coordinates-only model had dramatically lower diversity as measured by this metric (Figure 1, right). These findings suggest that modelling C$\alpha$ positions alone produces less biochemically plausible, more homogeneous structures, and furthermore, that measuring

---

[1]Generally, our structures had much higher RMSD than similar models (Luo et al., 2022). We suspect this occurs for many reasons, including not conditioning on scaffold information, pretraining on a more diverse dataset, and excluding antibodies with >90% sequence similarity.

RMSD to the ground truth structure fails to capture salient features of generated CDRs. We include examples of generated structures for both models in Figure 4 and in the Appendix.

### 3.2. Exploring how Diffusion Schedules Affect Performance

Score-based generative models of frames are unusual because noise must be applied over both the orientation and coordinate of each residue. These two processes introduce complexity because the relative position between a residue and its neighbors informs what orientations are possible and vice versa. Variance schedules have been shown to have a large impact on diffusion models for image generation (Karras et al., 2022); however, to our knowledge, there is almost no published data on how different noising schedules for translations and rotations affect frame generative models for proteins.

RMSD to the ground truth structure is the standard metric for evaluating the quality of generated CDR structures (Luo et al., 2022; Xie et al., 2023). We found that models with different variance schedules showed minimal variation in ground truth RMSD but varied significantly in their ability to generate physicochemically plausible structures (Table 1), where we defined structural violations using the violation loss functions from AlphaFold2 (Jumper et al., 2021). The quadratic translation schedule combined with a logarithmic schedule for rotations exhibited the lowest rate of structural violations, with >90% of generated structures being physicochemically plausible. Despite showing poor correspondence between generated and ground truth CDR structures, the generated loops still satisfy key criteria as both valid loops and potential binders, obeying the correct dihedral angle distribution at the correct distance from the epitope (Figure 2). These findings suggest that RMSD may not be an appropriate metric for assessing generative models of CDR loops, which have notably high structural diversity.

### 3.3. Evaluating Generative Models for Binders

Given that RMSD to the ground truth structure appears to have many limitations as a metric in the generative setting, we searched for additional metrics of the model's capacity to generate protein binders with high affinity and specificity. We reasoned that a key desideratum of any protein binder generative model is that the distribution of generated structures should be dependent on the target protein, an important evaluation that has thus far been largely neglected in the literature on deep learning-based protein binder design. We reasoned that the target-dependence of the model's generated CDR structures could be evaluated by comparing structures generated for epitopes transformed under different perturbation schemes. First, we generated a set of 10

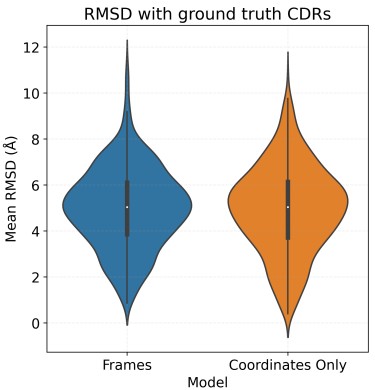
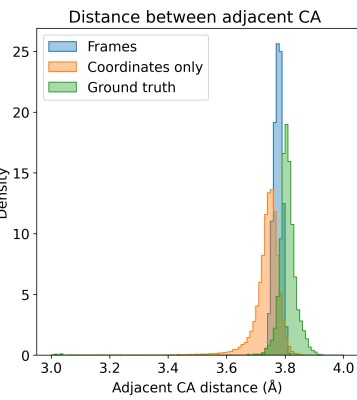
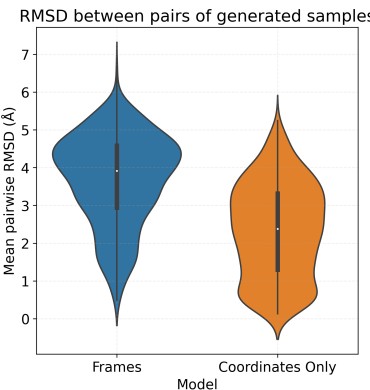

*Figure 1.* Rotational information improves generative models of binding peptides. From left to right: a comparison of the generated Cα RMSD to the ground truth, a histogram of the distances between adjacent Cα, and a comparison of the pairwise Cα RMSD values between 10 generated structures for each test set epitope (higher values indicate greater structural diversity).

*Table 1.* Choice of Variance Schedule Dramatically Affects Structure Quality

| Trans. Sched. | Rot. Sched. | RMSD (Å) | Internal Clashes (%) | Bond Length (%) | Bond Angle (%) | Epi.-CDR Clash (%) | Any Struct. Viol. (%) |
|---|---|---|---|---|---|---|---|
| Lin. | Log. | $4.98 \pm 2.14$ | 0.3 | 20.6 | 3.9 | 3.3 | 22.4 |
| Quad. | Log. | $4.93 \pm 2.15$ | 0.0 | 6.3 | 0.6 | 2.7 | 8.7 |
| Log. | Log. | $4.85 \pm 2.08$ | 88.9 | 96.2 | 43.9 | 5.7 | 97.1 |
| Sig. | Log. | $4.93 \pm 2.18$ | 0.6 | 6.1 | 1.8 | 3.4 | 9.2 |
| Lin. | Lin. | $5.04 \pm 2.08$ | 1.0 | 29.4 | 4.0 | 3.4 | 30.9 |
| Lin. | Quad. | $5.13 \pm 2.17$ | 0.8 | 18.6 | 1.7 | 3.5 | 20.5 |

CDRs against each test set CDR's original epitope, which we refer to as the WT epitope. Then, we randomly permuted epitopes in the test set, rotating and translating each random epitope to align with each WT epitope and re-generated ten CDRs of the same length as the WT epitope's native CDR. Similarly, to isolate the effect of the epitope's sequence, we permuted residue identities in the WT epitope - while keeping its structure constant - and generated a further 10 CDRs, a procedure we refer to as scrambling. Finally, we translated the WT epitope 20 Å away from the WT CDR's centre of mass (placed at the origin) - effectively removing any physical interactions between the target and CDR - and generated 10 CDRs. For each perturbation, we computed the mpRMSD between each of these sets of generated CDRs and the set of CDRs generated for the wild-type (original) epitope. Larger mpRMSDs indicate greater structural differences between sets of generated CDRs. Figure 3 shows that all three perturbations result in a notable increase in mpRMSD with CDR structures generated for the WT epitope compared to other CDR structures generated for the perturbed epitope, indicating that the generated structures are dependent on the WT epitope's structure, sequence, and positioning. They are even self-consistent within each set of perturbed epitopes, despite the perturbation generating synthetic CDR/epitope complexes. We also verified that the epitope perturbations generally had minimal effect on the

physicochemical plausibility of the generated CDRs (see Appendix).

## 4. Discussion and Conclusion

While SGMs for protein structure generation have catalysed great advances in *de novo* protein design, relatively little is known about the key components affecting their performance, and even less so in the context of designing binding proteins. Here we introduce LoopGen, a generative model and evaluation pipeline that designs CDR loop structures and enables direct comparison of various hyperparameter choices in the SGM setting. We use LoopGen to conduct, to our knowledge, the first direct comparison between modelling entire residue frames versus Cα coordinates alone. We hypothesised that modelling frames would provide specific benefits in designing binding peptides, which have highly variable structures and therefore significant degrees of freedom in their backbone dihedral angles. Indeed, results show that the Cα traces from structures generated using frame diffusion are significantly more diverse than structures generated solely using diffusion over Cα coordinates. We also compare models trained under different variance schedules and show that, while schedules do not have a significant effect on the commonly used ground truth RMSD metric, they do significantly influence the rate of

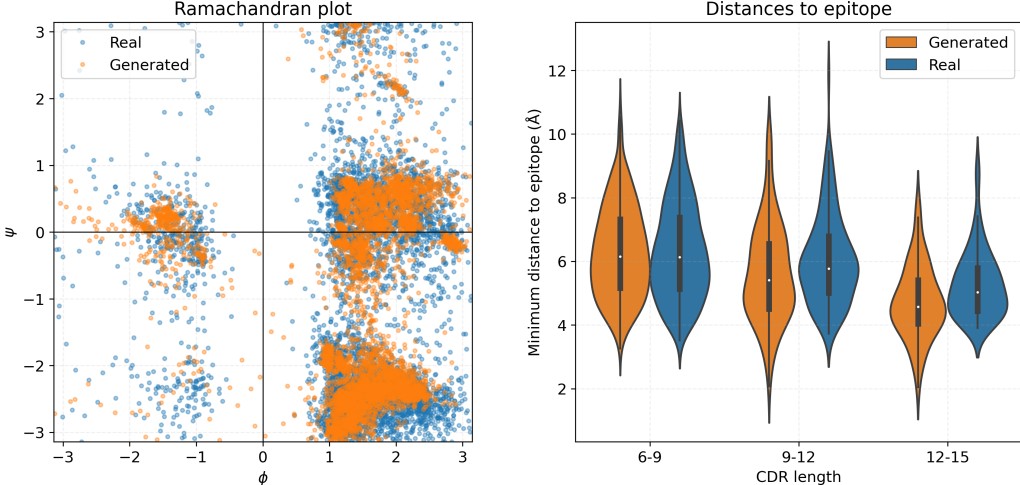

*Figure 2.* Left: Ramachandran Distribution of the generated CDRs compared to ground truth. Right: The minimum distance between an Cα on the CDR and the epitope for the ground truth and generated CDRs. Stratification by length reveals better performance on shorter CDRs.

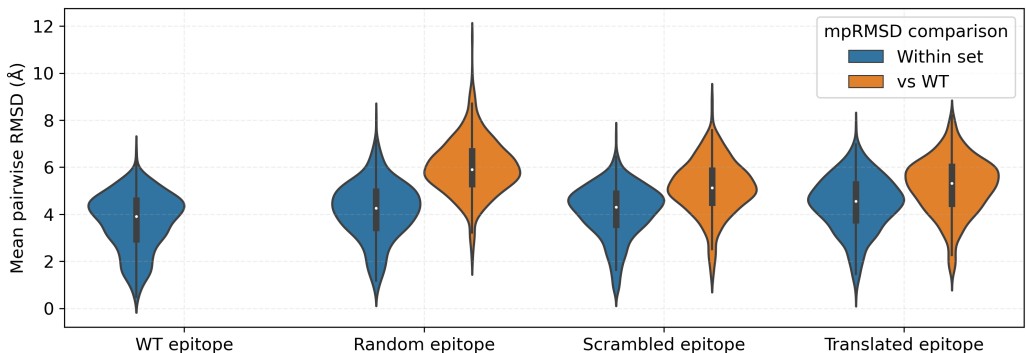

*Figure 3.* Analyzing the dependence of the generated CDR structures on the epitope. A set of ten CDR structures is generated for each of 687 test set epitopes (wild-type (WT) epitopes). Another ten are generated under each of three types of perturbation to each epitope: permutation (swapping the WT epitope with a random one from test set), scrambling (permuting residue identities within the WT epitope structure), and translation (translating the epitope 20 Å away from the CDR). Mean pairwise RMSDs are calculated within the CDRs generated for each epitope condition (blue) and between the set of CDRs generated for the WT epitope and permuted epitope (orange).

structural violations. Interestingly, similar observations of a discordance between RMSD and physicochemical plausibility have also been made in the small molecule generative model literature (Buttenschoen et al., 2023; Harris et al., 2023). The combination of a quadratic schedule for translations and a logarithmic schedule for rotations produced the optimal configuration for generating physicochemically plausible structures (Table 1), and we note that this combination corresponds to the greatest difference in variance growth rates between the rotational and translational noising processes (Figure 6, Appendix). A notable property of this difference is that the Cα coordinates of the loop are denoised before the residue orientations in the reverse diffusion process, a feature that aligns with biochemical intuition, since the Cα atoms act as the scaffold through which the

backbone dihedral angles (determined by the residue orientations) determine the final loop conformation. We also introduce novel metrics to assess generated binder structures by evaluating the extent to which they are dependent on the epitope (Figure 3). These *in silico* metrics are easy to compute and provide a crucial check that the model is using information from the target protein in the generative process, a key prerequisite for an effective binder design model.

Our experiments identify multiple promising avenues for further investigation. First, it should be noted that Loop-Gen requires, as input, a centre-of-mass for the designed CDR backbone; hence, as a generative model, it is currently most suitable for CDR backbone re-design, using an existing CDR structure as input. However, we have observed

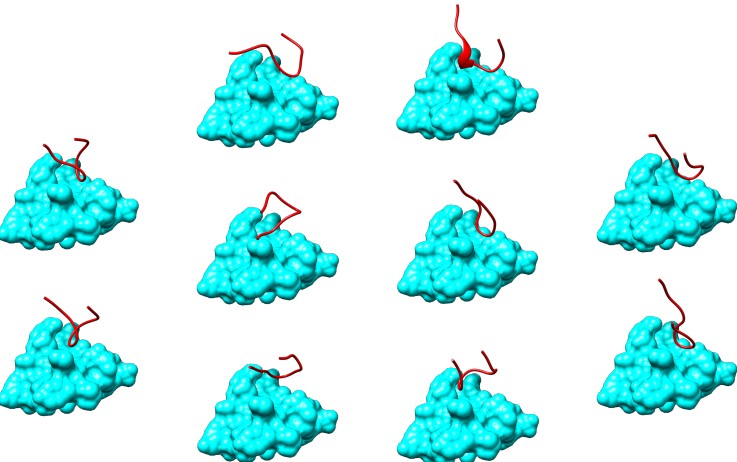

*Figure 4.* Ten generated CDR loops for a test set H-CDR3 epitope (PDB ID: 3ULU) using LoopGen. Columns are organised based on the qualitative structural features of the generated loop (from left to right: twist motif in the pocket, lateral binding interface, perpendicular binding interface, kink in the pocket). These structures highlight the model's capacity to generated diverse loop conformations.

promising results conditioning LoopGen's generative process on a centre-of-mass *generated* by another (standard euclidean) SGM as a single coordinate in $\mathbb{R}^3$ (unpublished data). Furthermore, while we study the incorporation of orientational information into the generative model, we only do so for the IGSO(3) diffusion framework (Yim et al., 2023; Watson et al., 2023). However, future research may benefit from exploring different forms of generative models over rotations (e.g. Anand & Achim (2022); Lin & AlQuraishi (2023)). Furthermore, although we showed significant performance variation across both translation and rotation variance schedules separately, a more extensive evaluation of the entire space of schedule combinations may identify configurations that yield even better results. Finally, we emphasise that our model alone is not sufficient for *de novo* CDR design because it only generates CDR structures, independent of sequences. While most methods to-date have relied on an inverse-folding model to annotate generated backbones (Ingraham et al., 2023; Watson et al., 2023; Bennett et al., 2024), previous work has also tested conducting structure and sequence generation concurrently (Luo et al., 2022). Although designing structure and sequence separately has been successful for large proteins, peptide binding is highly sensitive to the position and orientation of individual residues, and hence future research may benefit from comparing methods for post-hoc sequence design to incorporating sequence generation directly into the SGM framework. Finally, our experiments are limited to CDRs. Our findings about metrics and variance schedules should be independently replicated on datasets of whole proteins.

To facilitate further research into these questions and others, we are releasing our code, model weights, and our cleaned training/evaluation dataset from SAbDab (Dunbar et al.,

2014) that has been filtered to remove repetitive sequences and truncated to individual CDR and epitope fragments. The model can be used with its pre-trained weights to generate CDR loops for input epitope PDB files or re-trained *ab initio* with other binding peptide datasets. We hope that our findings and the accompanying materials guide the development of improved generative models for binding peptides and proteins more broadly.

## Acknowledgements

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
