## A  LOOPGEN TECHNICAL DETAILS

Score-based generative modelling via stochastic differential equations (SDEs) (Song et al., 2021) models the process of adding noise to data with the following forward SDE:

$$\mathrm{d}\mathbf{x} = f(\mathbf{x}, t)\mathrm{d}t + g(t)\mathrm{d}\mathbf{w} \tag{1}$$

Where $\mathbf{x}$ are the data and $\mathbf{w}$ is a Brownian motion. The forward SDE has a corresponding reverse SDE, which models how data are generated from noise:

$$\mathrm{d}\mathbf{x} = \left[ f(\mathbf{x}, t) - g(t)^2 \nabla \log p_t(\mathbf{x}) \right] \mathrm{d}t + g(t)\mathrm{d}\bar{\mathbf{w}} \tag{2}$$

Where $\mathrm{d}t$ is an infinitesimal negative time step and $\bar{\mathbf{w}}$ is a Brownian motion for time moving in the reverse direction. The score $\nabla \log p_t(\mathbf{x})$ is approximated with an estimator $s_\theta(\mathbf{x}, t)$ by optimising a score-matching objective, typically of the form:

$$\mathcal{L}(\theta) = \mathbb{E}_{\mathbf{x},t} \left[ \lambda(t) \| \nabla \log p_t(\mathbf{x}) - s_\theta(\mathbf{x}, t) \|_2^2 \right] \tag{3}$$

Where the coefficients $\lambda(t)$ are usually chosen to balance the expected magnitudes of the ground truth scores, i.e. $\lambda(t) = 1/\mathbb{E}\left[ \| \nabla \log p_t(\mathbf{x}) \|^2 \right]$ (Song et al., 2021).

An important extension of SGM that has found applications in protein design is the *Riemannian SGM* framework, which extends the forward and reverse SDEs to Riemannian manifolds on which appropriate Brownian motion and probability densities can be defined (De Bortoli et al., 2022), including the manifold of three-dimensional proper rotations $SO(3)$. In proteins, the orientation of each amino acid can be represented using a three-dimensional rotation matrix (Jumper et al., 2021), and recent generative modelling approaches have attempted to generate novel protein structures by formulating SGM approaches for both the rotational and translational component of each residue (Watson et al., 2023; Yim et al., 2023). The combination of a rotation and a translation in this context is known as a frame and is a member of the Lie group $SE(3)$. Yim et al. (2023) formalised the theory of SGM for frames in their recent work in which they propose a protein generative modelling approach known as FrameDiff.

To define generative models on $SE(3)$, Yim et al. (2023) note that $SE(3)$ may be identified as $SO(3) \times \mathbb{R}^3$ (Yim et al., 2023) under a proper choice of inner product, allowing diffusion processes over translations ($\mathbb{R}^3$) and rotations ($SO(3)$) to be performed separately. Let $\mathbf{T}^{(t)} = (\mathbf{R}^{(t)}, \mathbf{X}^{(t)})$, where $\mathbf{X}^{(t)} \in \mathbb{R}^3$ represents a translation and $\mathbf{R}^{(t)} \in SO(3)$ represents a rotation at time $t$. Let $\mathbf{B}_\mathcal{M}^{(t)}$ represent Brownian motion on a manifold $\mathcal{M}$.[1] Then, the forward diffusion process can be described as follows:

$$\mathrm{d}\mathbf{T}^{(t)} = [0, -\frac{1}{2}\mathrm{P}\mathbf{X}^{(t)}]\mathrm{d}t + \left[ \mathrm{d}\mathbf{B}_{SO(3)}^{(t)}, \mathrm{d}\mathrm{P}\mathbf{B}_{\mathbb{R}^3}^{(t)} \right] \tag{4}$$

where $\mathrm{P}$ is a projection matrix centering the translations at the origin. Removing the center of mass maintains SE(3)-invariance. Notably, there is no drift term for the rotations $\mathbf{R}^{(t)}$. Similarly, the reverse process is defined as

$$\begin{aligned}
\mathrm{d}\mathbf{R}^{(t)} &= \nabla_\mathbf{R} \log p_t(\mathbf{T}^{(t)})\mathrm{d}t + \mathrm{d}\mathbf{B}_{SO(3)}^{(t)} \\
\mathrm{d}\mathbf{X}^{(t)} &= \mathrm{P}\left( \frac{1}{2}\mathbf{X}^{(t)} + \nabla_\mathbf{x} \log p_t(\mathbf{T}^{(t)}) \right) \mathrm{d}t + \mathrm{Pd}\mathbf{B}_{\mathbb{R}^3}^{(t)}
\end{aligned} \tag{5}$$

LoopGen takes inspiration from FrameDiff (Yim et al., 2023) and focuses specifically on adapting their proposed SGM framework to make score-based generative models for frames more flexible and more straightforward to train. FrameDiff relies on the Invariant Point Attention (IPA) architecture

---

[1]Please see De Bortoli et al. (2022) and section 2 of Yim et al. (2023) for more details on deriving the forward and reverse processes on $\mathcal{M}$.

(Jumper et al., 2021) to predict the ground truth frame $\mathbf{T}^{(0)} = (\mathbf{R}^{(0)}, \mathbf{x}^{(0)})$ for each residue, and calculates the predicted score at each time step by backpropagating through the score function:

$$s_\theta(\mathbf{T}^{(t)}, t) = \nabla \log p_t(\mathbf{T}^{(t)} | \hat{\mathbf{T}}^{(0)}) \tag{6}$$

$$= \left[ \nabla \log p_t(\mathbf{R}^{(t)} | \hat{\mathbf{R}}^{(0)}), \nabla \log p_t(\mathbf{x}^{(t)} | \hat{\mathbf{x}}^{(0)}) \right] \tag{7}$$

where the score for a frame is defined separately for rotations and translations. Importantly, this formulation restricts the form of the estimator $s_\theta(\mathbf{T}^{(t)}, t)$ to functions that directly predict frames $\hat{\mathbf{T}}^{(0)} = (\hat{\mathbf{R}}^{(0)}, \hat{\mathbf{x}}^{(0)})$. However, we note that for translations $\mathbf{x} \in \mathbb{R}^3$, the score $\nabla \log p_t(\mathbf{x})$ for a Gaussian density can be predicted directly as an SO(3)-equivariant 3-vector, and likewise for rotations $\mathbf{R} \in SO(3)$, the isomorphism between the Lie algebra $\mathfrak{so}(3)$ and $\mathbb{R}^3$ can be exploited to represent and predict the rotational score $\nabla \log p_t(\mathbf{R})$ directly as an element of $\mathbb{R}^3$. For the IGSO(3) distribution (Leach et al., 2022), we note that this formulation has the key advantage of not requiring continuous re-estimation of the infinite sum in the IGSO(3) density function (8) during the training process, which can require significant compute and numerical instabilities. Instead, scores can be computed before training for a discretized support (on the interval $[0, 2\pi]$), cached, and returned as constants during training since backpropagation through the score function is no longer required. Furthermore, the reparameterization of the scores as elements of $\mathbb{R}^3$ allows a much wider variety of estimator architectures to be used, beyond functions that directly output frames.

## A.1 REPRESENTING THE IGSO(3) SCORE

The IGSO(3) distribution is defined over axis-angle parameterizations of three-dimensional rotations. These have the form $\omega\hat{\theta}$, where $\hat{\theta}$ is a unit-length axis of rotation and $\omega$ is the angle of rotation. The density is uniform over axes $\hat{\theta}$, and the density over angles of rotation $p(\omega)$ has the following form:

$$f(\omega) = \sum_{l=0}^{\infty} (2l + 1) \exp\left[-l(l+1)\sigma^2\right] \frac{\sin\left(\left[l + \frac{1}{2}\right]\omega\right)}{\sin\left(\frac{\omega}{2}\right)} \tag{8}$$

$$p(\omega) = \frac{1 - \cos(\omega)}{\pi} f(\omega) \tag{9}$$

where $\sigma^2$ is a variance parameter and the normalising constant $\frac{1-\cos\omega}{\pi}$ ensures that the joint distribution over all axes and angles integrates to 1 (Leach et al., 2022). Adding noise at time $t$ to a sample rotation $\mathbf{R}^{(0)}$ then involves sampling an *incremental* rotation from the corresponding time step distribution parameterised by variance $\varsigma_t^2$, and then right multiplying with the sample rotation to obtain a sample from $p(\mathbf{R}^{(t)} | \mathbf{R}^{(0)})$:

$$\mathcal{R} \sim \text{IGSO3}(\mu = \mathbf{I}, \sigma^2 = \varsigma_t^2) \tag{10}$$

$$\mathbf{R}^{(t)} = \mathcal{R}\mathbf{R}^{(0)} \tag{11}$$

The score of the noised data distribution $p(\mathbf{R}^{(t)} | \mathbf{R}^{(0)})$ then has the following form:

$$\nabla \log p(\mathbf{R}^{(t)} | \mathbf{R}^{(0)}) = \frac{\text{Log}\left([\mathbf{R}^{(0)}]^\top \mathbf{R}^{(t)}\right)}{\omega\left([\mathbf{R}^{(0)}]^\top \mathbf{R}^{(t)}\right)} \frac{f'\left(\omega([\mathbf{R}^{(0)}]^\top \mathbf{R}^{(t)})\right)}{f\left(\omega([\mathbf{R}^{(0)}]^\top \mathbf{R}^{(t)})\right)} \tag{12}$$

Where we use $\omega(\mathbf{R})$ to denote the angular component $\omega$ of an axis-angle representation $\omega\hat{\theta}$ of a rotation matrix $\mathbf{R}$, and follow the convention of Solà et al. (2021), using Log to indicate the logarithmic matrix map from a Lie group to Euclidean space, i.e. $\text{Log} : SO(3) \to \mathbb{R}^3$ here. Note that we do not follow previous works (Yim et al., 2023; Watson et al., 2023) which represent $\nabla \log p(\mathbf{R}^{(t)} | \mathbf{R}^{(0)})$ as an element of the tangent space at the noised rotation $\mathcal{T}_{\mathbf{R}^{(t)}} = \{\mathbf{R}^{(t)}\Theta : \Theta \in \mathfrak{so}(3)\}$; instead,

the score is always represented at the tangent space at the identity rotation (i.e. $\mathfrak{so}(3)$), hence no further processing is needed before passing predicted scores through the exponential map to update rotations in the generation process. We see that $\nabla \log p(\mathbf{R}^{(t)}|\mathbf{R}^{(0)})$ in this representation is SO(3)-invariant since $(\mathbf{R}^{(0)})^{\top}\mathbf{R}^{(t)}$ is SO(3)-invariant.

## A.2 Training details

To stabilise the score-matching loss for rotations we use coefficients:

$$\lambda(t) = 1/\mathbb{E}\left[\|\nabla \log p_t(\mathbf{R}^{(t)}|\mathbf{R}^{(0)})\|^2\right]$$

that we estimate via a Monte Carlo procedure, calculating the empirical mean norm of the scores for each time point over 50,000 uniformly-sampled rotations $\mathbf{R}^{(0)}$. For translations, we found that scaling the ground truth score by a factor of $-\sigma_t$ - where $\sigma_t = \sqrt{\text{Var}[\mathbf{x}^{(t)}|\mathbf{x}^{(0)}]}$ is the standard deviation of noised translations at time $t$ - stabilised training significantly, noting that:

$$
\begin{aligned}
-\sigma_t \nabla \log p_t(\mathbf{x}^{(t)}|\mathbf{x}^{(0)}) &= -\sigma_t \frac{\mu - \mathbf{x}^{(t)}}{\sigma_t^2} \\
&= \frac{\mathbf{x}^{(t)} - \mu}{\sigma_t} \\
&\sim \mathcal{N}(0, \mathbf{I})
\end{aligned}
\tag{13}
$$

Which corresponds to the standard diffusion model training objective (Ho et al., 2020). A single training step involving sampling and loss calculation is shown below (Algorithm 1) for an input set of frames $\{(\mathbf{R}_i^{(0)}, \mathbf{x}_i^{(0)})\}_{i=1}^N$, representation of the epitope $\mathbf{Z}$, score estimator $s_\theta$, rotation and translation coefficient schedules $\beta^r$ and $\beta^x$ (from which variances are calculated), rotation score matching coefficient $\lambda^r$, and time step $t \in (0, 1)$. Integrals are all approximated using the trapezoid rule. Our noising regime corresponds to the variance-preserving SDE from Song et al. (2021).

---

**Algorithm 1** LoopGen Training Step

---

**Require:** $\{(\mathbf{R}_i^{(0)}, \mathbf{x}_i^{(0)})\}_{i=1}^N, \mathbf{Z}, s_\theta, \beta^r, \beta^x, \lambda^r, t$

$\quad \varsigma_t^2 \leftarrow 1 - \exp\left(-\int_0^t \beta^r(s)\, \mathrm{d}s\right)$         $\triangleright \text{Var}[\mathbf{R}^{(t)}|\mathbf{R}^{(0)}]$

$\quad \sigma_t^2 \leftarrow 1 - \exp\left(-\int_0^t \beta^x(s)\, \mathrm{d}s\right)$         $\triangleright \text{Var}[\mathbf{x}^{(t)}|\mathbf{x}^{(0)}]$

$\quad$**for** i = 1, ..., N **do**

$\quad\quad\quad \mathcal{R} \sim \text{IGSO3}(\mu = \mathbf{I}, \sigma^2 = \varsigma_t^2)$        $\triangleright$ Sample rotational noise

$\quad\quad\quad \mathbf{R}_i^{(t)} \leftarrow \mathcal{R}\mathbf{R}_i^{(0)}$            $\triangleright$ Get noised rotation

$\quad\quad\quad \mathbf{x}_i^{(t)} \sim \mathcal{N}(\mu = \mathbf{x}_i^{(0)} \exp\left(-\frac{1}{2}\int_0^t \beta^x(s)\, \mathrm{d}s\right), \sigma^2 = \sigma_t^2)$    $\triangleright$ Sample noised translation

$\quad$**end for**

$\quad \{(\hat{\mathbf{y}}_i^r, \hat{\mathbf{y}}_i^x)\}_{i=1}^N \leftarrow s_\theta(\{(\mathbf{R}_i^{(t)}, \mathbf{x}_i^{(t)})\}_{i=1}^N, \mathbf{Z}, t)$    $\triangleright$ Predict scores for each residue

$\quad \mathcal{L}^r \leftarrow \lambda^r(t) \sum_{i=1}^N \|\nabla \log p_t(\mathbf{R}_i^{(t)}|\mathbf{R}_i^{(0)}) - \hat{\mathbf{y}}_i^r\|^2$    $\triangleright$ Rotation score loss

$\quad \mathcal{L}^x \leftarrow \sum_{i=1}^N \|-\sigma_t \nabla \log p_t(\mathbf{x}_i^{(t)}|\mathbf{x}_i^{(0)}) - \hat{\mathbf{y}}_i^x\|^2$    $\triangleright$ Translation score loss

$\quad \mathcal{L} \leftarrow \mathcal{L}^r + \mathcal{L}^x$              $\triangleright$ Combined loss

$\quad$**return** $\mathcal{L}$

---

For generation (18), we broadly follow the procedures outlined by Ho et al. (2020) for translations and Yim et al. (2023) for rotations, using the notation Exp to denote the exponential map from euclidean space to a Lie group, where $\text{Exp} : \mathbb{R}^3 \to SO(3)$ in this case. We found empirically that scaling the score term in the rotational update $\gamma g_t^2 \hat{\mathbf{y}}_i^r$ by a factor of 2 improved the quality of

generated structures. For our experiments we used number of time steps $T = 100$ and a noise scaling term of $\zeta = 0.2$. We found that reasonable structures were generated for values up to $\zeta = 0.5$.

---

**Algorithm 2** LoopGen Generation

---

**Require:** $\mathbf{Z}, s_\theta, \beta^r, \beta^x, \zeta, T, N$

$\quad \gamma \leftarrow \frac{1}{T}$

$\quad$ **for** i = 1, ... , N **do** $\qquad\qquad\qquad\qquad\qquad\qquad\qquad\qquad$ ▷ Sample frames for N residues

$\qquad$ Sample $\mathbf{R}_i^{(1)} \sim \text{UniformSO3}$

$\qquad$ Sample $\mathbf{x}_i^{(1)} \sim \mathcal{N}(0, \mathbf{I})$

$\quad$ **end for**

$\quad$ **for** $t = 1 - \gamma, 1 - 2\gamma, ..., 0$ **do**

$\qquad g_t \leftarrow \sqrt{\beta^r(t)} \qquad\qquad\qquad\qquad\qquad\qquad\qquad\qquad$ ▷ Diffusion coefficient for VP-SDE

$\qquad \alpha_t \leftarrow 1 - \exp\left(-\int_t^{t+\gamma} \beta^x(s)\, \mathrm{d}s\right) \qquad\qquad\qquad$ ▷ $\text{Var}[\mathbf{x}^{(t+\gamma)}|\mathbf{x}^{(t)}]$

$\qquad \sigma_t^2 \leftarrow 1 - \exp\left(-\int_0^t \beta^x(s)\, \mathrm{d}s\right) \qquad\qquad\qquad$ ▷ $\text{Var}[\mathbf{x}^{(t)}|\mathbf{x}^{(0)}]$

$\qquad \{(\hat{\mathbf{y}}_i^r, \hat{\mathbf{y}}_i^x)\}_{i=1}^N \leftarrow s_\theta(\{(\mathbf{R}_i^{(t+\gamma)}, \mathbf{x}_i^{(t+\gamma)})\}_{i=1}^N, \mathbf{Z}, t) \quad$ ▷ Predict scores for each residue

$\qquad$ **for** i = 1, ... N **do**

$\qquad\qquad$ Sample $\epsilon^x \sim \mathcal{N}(0, \mathbf{I}) \qquad\qquad\qquad\qquad\qquad$ ▷ Noise for translations

$\qquad\qquad$ Sample $\epsilon^r \sim \mathcal{N}(0, \mathbf{I}) \qquad\qquad\qquad\qquad\qquad$ ▷ Noise for rotations

$\qquad\qquad \mathbf{x}_i^{(t)} \leftarrow \frac{\mathbf{x}_i^{(t+\gamma)} - \alpha_t}{\sigma_t \sqrt{1-\alpha_t}} \hat{\mathbf{y}}_i^x + \zeta\sqrt{\alpha_t}\epsilon^x \qquad\qquad$ ▷ Translation update

$\qquad\qquad \mathbf{R}_i^{(t)} \leftarrow \mathbf{R}_i^{(t+\gamma)} \text{Exp}(2\gamma g_t^2 \hat{\mathbf{y}}_i^r + \zeta\sqrt{\gamma} g_t \epsilon^r) \qquad\qquad$ ▷ Rotation update

$\qquad$ **end for**

$\quad$ **end for**

$\quad$ **return** $\{(\mathbf{R}_i^{(0)}, \mathbf{x}_i^{(0)})\}_{i=1}^N$

---

### A.3 Tests for epitope dependence

Here we outline the procedures for evaluating the conditionality of generated structures on the epitope. The fundamental principle we use is that dependence on the epitope can be measured by estimating similarity between sets of structures generated under different perturbations of the epitope. Let $\mathbf{X}$ be a random variable representing a generated structure and $\mathbf{Y}$ another generated structure. The quantity of interest is then the expected RMSD between $\mathbf{X}$ and $\mathbf{Y}$, i.e. the mean pairwise RMSD (mpRMSD):

$$\text{mpRMSD} = \mathbb{E}_{\mathbf{X}}[\mathbb{E}_{\mathbf{Y}}[\text{RMSD}(\mathbf{X}, \mathbf{Y})]] \tag{14}$$

For a given epitope $\mathbf{Z}$ and a perturbed version of that epitope $\mathbf{Z}'$, we compare the mpRMSD for samples drawn from $p_\theta(\mathbf{X}|\mathbf{Z})$ and $p_\theta(\mathbf{Y}|\mathbf{Z}')$ to those generated under identical epitope conditions, i.e. $p_\theta(\mathbf{X}|\mathbf{Z})$ and $p_\theta(\mathbf{Y}|\mathbf{Z})$. For a given set of structures generated for a WT epitope $\mathbf{Z}$, with sets of residue coordinates $\left\{\{\mathbf{x}_1^{(i)}, ..., \mathbf{x}_N^{(i)}\}\right\}_{i=1}^M$, the mpRMSD *within* the set of structures is estimated as:

$$\text{mpRMSD} \approx \frac{2}{M(M-1)} \sum_{i=1}^M \sum_{j=i+1}^M \sqrt{\frac{1}{N}\sum_{n=1}^N \|\mathbf{x}_n^{(i)} - \mathbf{x}_n^{(j)}\|^2} \tag{15}$$

Where the scaling factor is the reciprocal of $M$ choose 2, calculating the mean RMSD over all possible unique pairings of structures in the set. We use the within-group mpRMSD to measure

the expected variation between CDR loop structures generated for a constant epitope ("WT epitope vs WT epitope", figure **??**). The mpRMSD *between* different sets of structures - where one set $\left\{\{\mathbf{x}_1^{(i)}, ..., \mathbf{x}_N^{(i)}\}\right\}_{i=1}^M$ is generated for the WT epitope $\mathbf{Z}$ and another set $\left\{\{\mathbf{x'}_1^{(j)}, ..., \mathbf{x'}_N^{(j)}\}\right\}_{j=1}^K$ is generated for the perturbed epitope $\mathbf{Z}'$ - is similarly estimated:

$$\text{mpRMSD} \approx \frac{1}{MK} \sum_{i=1}^M \sum_{j=1}^K \sqrt{\frac{1}{N} \sum_{n=1}^N \|\mathbf{x}_n^{(i)} - \mathbf{x'}_n^{(j)}\|^2} \tag{16}$$

We compare the within-group mpRMSD values for structures generated on the WT epitope to the mpRMSD values between structures generated for the WT epitope and for epitopes perturbed using three methods: permutation (swapping with another random epitope), sequence scrambling, and translation. For permutation, to ensure that global orientation and position do not contribute to differences in mpRMSD, we perform an alignment of a randomly-sampled epitope from the test set with the WT epitope by aligning their centre of masses and the CDR-epitope centroid difference vectors. The CDR is generated of the same size as the WT CDR for RMSD comparison. Epitope scrambling was performed by shuffling the residue sequence labels of each epitope, setting $C\beta$ coordinates of residues whose residue identity becomes glycine to their $C\alpha$ coordinates, preventing information leakage (glycines are represented the same way during training), while $C\beta$ coordinates for glycines that convert into non-glycine residues were imputed with tetrahedral geometry. Epitope translation is performed by translating all epitope atoms 20 Å along the axis connecting the epitope centroid to the CDR centroid. Translation is performed along this axis in the direction opposite to the CDR centroid's position relative to the epitope centroid.

## B    ADDITIONAL DATA ON EPITOPE PERTURBATION EXPERIMENT

Table 1: Structural violation rates of generated CDRs under different epitope perturbations

| Perturbation | Internal Clashes (%) | Bond Length (%) | Bond Angle (%) | Epi.-CDR Clash (%) | Any Struct. Viol. (%) |
|---|---|---|---|---|---|
| WT (baseline) | 0.0 | 6.3 | 0.6 | 2.7 | 8.7 |
| Permuted | 0.1 | 8.1 | 1.0 | 23.1 | 28.5 |
| Scrambled | 0.0 | 7.1 | 0.8 | 2.5 | 9.3 |
| Translated | 0.0 | 6.2 | 0.6 | 0.0 | 6.3 |

Table 1 describes structural violation statistics for CDRs generated after perturbing the epitope via permutation, sequence scrambling, and translation away from the CDR center of mass. Sequence scrambling and translation of the epitope induced no significant effects on the physicochemical plausibility of generated structures. However, replacement of the WT epitope with a random epitope was accompanied by a notable increase in the rate of steric clashes between the CDR and epitope. Since the random epitope was rotationally aligned and placed at the center of mass of the WT epitope before CDR generation, we hypothesise that these clashes are driven by permuted epitopes which are significantly larger or deviate in shape from the WT epitope, increasing the probability of physical clashes with the CDR. Nevertheless, we note that even under epitope permutation, the violation statistics intrinsic to the generated CDR (bond length, bond angle, and internal clashes) do not deviate significantly from the results observed under WT generation. Therefore, these experiments confirm that the model's generated structures are robust to different perturbations of the epitope and provide reassuring evidence of its capacity to generalise to out-of-distribution epitopes.

## C  EXAMPLES OF STRUCTURES GENERATED WITH A COORDINATES-ONLY MODEL

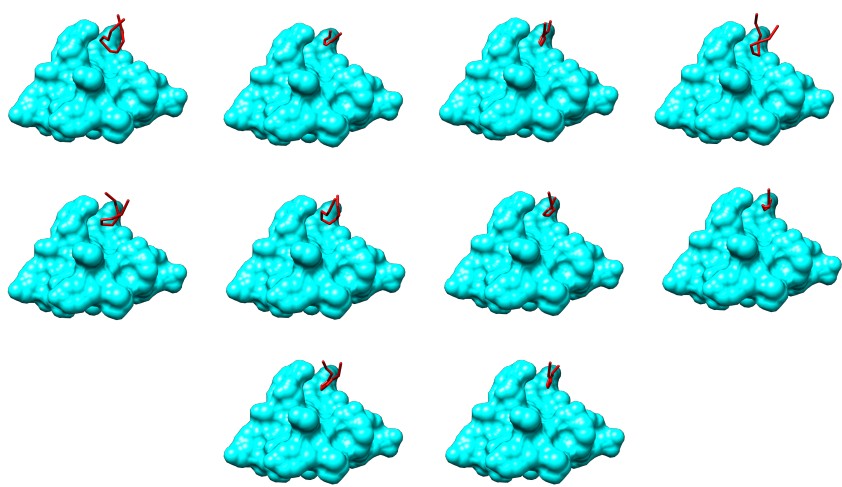

Figure 1: 10 generated CDR loops for a test set epitope (PDB ID: 3ULU) using LoopGen with a diffusion model over C$\alpha$ coordinates only (only C$\alpha$ coordinates are shown).

## D  DIFFUSION VARIANCE SCHEDULES

We test four different variance schedules parameterised by coefficients $\{\beta(t)\}_{t \in [0,1]}$ and hyperparameters $\beta_{\max}$ and $\beta_{\min}$. These coefficients are calculated as follows, for each schedule:

$$\beta(t) = t(\beta_{\max} - \beta_{\min}) \qquad \text{Linear} \qquad (17)$$

$$\beta(t) = \log\left(t(\exp[\beta_{\max}]) + (1-t)\exp[\beta_{\min}]\right) \qquad \text{Logarithmic} \qquad (18)$$

$$\beta(t) = \left[t\left(\sqrt{\beta_{\max}} - \sqrt{\beta_{\min}}\right)\right]^2 \qquad \text{Quadratic} \qquad (19)$$

$$\beta(t) = \frac{\beta_{\max} - \beta_{\min}}{1 + \exp\left[-(2t-1)x_{\max}\right]} \qquad \text{Sigmoid} \qquad (20)$$

$$(21)$$

Where in the sigmoid schedule, $2t - 1 \in [-1, 1]$ is a rescaled version of the timestep $t \in [0, 1]$, and $x_{\max}$ is a hyperparameter chosen to have a value of 6. For all schedules, values of $\beta_{\max} = 20$ and $\beta_{\min} = 10^{-4}$ were used for the translations and $\beta_{\max} = 1.5$ and $\beta_{\min} = 0.1$ for rotations (according to Yim et al. (2023)). The coefficients $\beta(t)$ are related to the variances of the forward process by the solution to the variance-preserving SDE (Song et al., 2021):

$$\text{Var}[\mathbf{x}^{(t)}|\mathbf{x}^{(0)}] = 1 - \exp\left(-\int_0^t \beta(s)\, \text{d}s\right) \qquad (22)$$

The values of $\text{Var}[\mathbf{x}^{(t)}|\mathbf{x}^{(0)}]$ are plotted by time step in Figure 2.

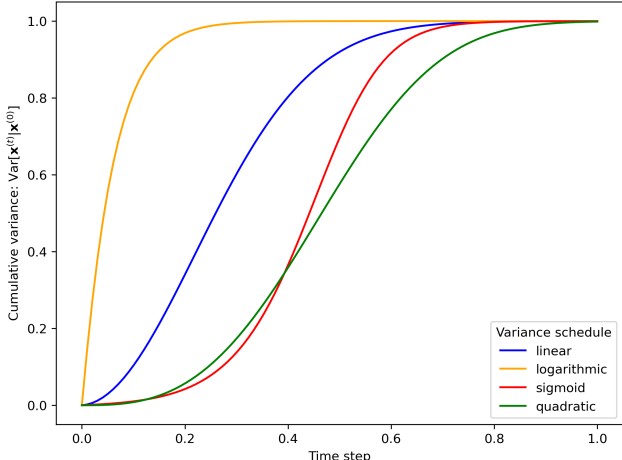

Figure 2: The cumulative variance $\mathrm{Var}[\mathbf{x}^{(t)}|\mathbf{x}^{(0)}]$ of the different variance schedules as a function of timestep $t \in [0,1]$. We find that using the slower schedules (quadratic and sigmoid) for noising the translations improves performance.