# OpenReview forum: "Score-Based Generative Models For Binding Peptide Backbones"
_ICML.cc/2024/Workshop/ML4LMS — ML4LMS Poster_

### Official Review · Reviewer_MJWW · 2024-06-10
**Paper evaluation design choices and metric of score-based generative models**

**Rating:** 7
**Confidence:** 4

**Review:**

Summary: The manuscript proposes LoopGen, a framework for score-based generative models to design short protein-binding peptide structures. In the papers, the authors examine the design of CDRH3 loops in antibodies specifically. They analyze three main points: (i) the representation used in the diffusion process; (ii) the effect of different diffusion schedules; and (iii) the current metrics used to evaluate performance.

Comments:

1. It would be also interesting to analyze the effect of a full-atom coordinate representation compared with the frame representation, especially regarding diversity metrics.
2. The reviewer suggests a mathematical explanation of the diffusion schedules used for analysis. Usually, these schedules have a strong theoretical mathematical formulation behind their modeling for specific applications such as 3D structure generation, in which the representation is very important. It would be interesting to have a deeper analysis on that.
3. Metrics: usual metrics to evaluate these systems involve refoldability with structure prediction networks. For the specific case of the setting used in the manuscript, it would be interesting to have epitope-conditioned metrics such as a separate structure-based scoring network to use for evaluation or to evaluate the scores given by the SGM. Specifically for the antigen-conditioned antibody design, it seems critical the need to have a separate in silico metric to score promising designs.

---

### Official Review · Reviewer_uYja · 2024-06-11

**Rating:** 5
**Confidence:** 3

**Review:**

The authors develop a GNN based diffusion model named LoopGen to determine CDR structure conditioned on a target epitope. The model predicts Ca positions and orientations of some predefined length but does not predict a CDR sequence. The authors present interesting comparisons of feature choices and diffusion schedules as well as novel evaluation metrics.

Pros:
- The authors provide a useful analysis of training on frames (which include CDR vector features + Ca coordinates) vs. Ca coordinates alone, showing that although RMSDs are comparable, the frame-based model maintains better structural integrity and diversity
- Noise schedule comparisons provide a useful guide as many approaches rely on applying noise to different manifolds (translation, rotation, torsions, etc)
- The permuted, scrambled, and translated epitopes provide additional validation of the model’s ability to generate diverse and physically plausible CDR structures despite out-of-sample conditioning

Cons:
- There is a systematic shift in adjacent Ca distance for both the frames and coordinates models (underpredicted compared the ground-truth). There is no explanation of this other than mentioning the frames performs better than coordinates, but this may reflect some bias in the model
- Although this particular exploration of the noise schedules for rotations vs. translation may be novel, there has been previous work in (Yim et. al 2023) and related papers establishing the benefits of combining a linear translation schedule with a logarithmic diffusion one. The authors results show further improvement by incorporating a quadratic schedule according to their metrics, but the results are quite similar to the baseline approach
- Because this model only predicts structure (inverse folding is required to obtain sequence) and CDRs are intrinsically disordered, it’s unclear how informative RMSD is as a metric.
- Although extensive hyperparameter comparisons are shown, there are no comparisons to other models, and therefore it is hard to evaluate how well an optimized LoopGen model is performing compared to the state-of-the-art
- All predictions are conditioned on the length of the CDR and the CDR center-of-mass. It is reasonable to provide the length a priori, however, it is unclear that the CDR center-of-mass would be known ahead of time. I could foresee this being the case when using a template that is known to have a very similar structure, but in this case the structure prediction task would be less important. Some discussion or ablations would be useful to show how the model performs or could be trained if no COM is provided.

Small points:
- In Table 1, an indication of which metrics should be maximized or minimized would be helpful
- In figure 1, violin plots of RMSD extend into negative regions. This is likely an artifact from the KDE, but plots should be clipped at zero (clip=0 in seaborn).
- A very similar version of this paper was published on Arxiv 6 months ago (found while researching background) along with a Github. While this does not violate the conference terms, it does somewhat reduce novelty and anonymity (eg. the name LoopGen could have been omitted from this work).

---

### Official Review · Reviewer_Vwzz · 2024-06-12
**-**

**Rating:** 8
**Confidence:** 3

**Review:**

-